# Multi-Step Internet Traffic Forecasting Models with Variable Forecast Horizons for Proactive Network Management [note 1]

**DOI:** 10.3390/s24061871

**Published:** 2024-03-14

**Authors:** Sajal Saha, Anwar Haque, Greg Sidebottom

**Affiliations:** 1Department of Computer Science, University of Northern British Columbia, Prince George, BC V2N 4Z9, Canada; 2Department of Computer Science, Western University, London, ON N6A 3K7, Canada; ahaque32@uwo.ca; 3Juniper Networks, Kanata, ON K2K 3E7, Canada; gsidebot@juniper.net

**Keywords:** anomaly detection, gradient boosting, gradient descent, traffic forecast, traffic prediction, machine learning

## Abstract

The ISP (Internet Service Provider) industry relies heavily on internet traffic forecasting (ITF) for long-term business strategy planning and proactive network management. Effective ITF frameworks are necessary to manage these networks and prevent network congestion and over-provisioning. This study introduces an ITF model designed for proactive network management. It innovatively combines outlier detection and mitigation techniques with advanced gradient descent and boosting algorithms, including Gradient Boosting Regressor (GBR), Extreme Gradient Boosting (XGB), Light Gradient Boosting Machine (LGB), CatBoost Regressor (CBR), and Stochastic Gradient Descent (SGD). In contrast to traditional methods that rely on synthetic datasets, our model addresses the problems caused by real aberrant ISP traffic data. We evaluated our model across varying forecast horizons—six, nine, and twelve steps—demonstrating its adaptability and superior predictive accuracy compared to traditional forecasting models. The integration of the outlier detection and mitigation module significantly enhances the model’s performance, ensuring robust and accurate predictions even in the presence of data volatility and anomalies. To guarantee that our suggested model works in real-world situations, our research is based on an extensive experimental setup that uses real internet traffic monitoring from high-speed ISP networks.

## 1. Introduction

Internet traffic volume forecasting is one of the most important tasks in the proactive management of modern telecommunication networks. Improving the accuracy and efficiency of traffic demand forecasting can help ISP companies develop reasonable business planning and enhance the industry’s economic benefits. Moreover, forecasting results with high accuracy can also be effective for better resource management, route scheduling, short- and long-term business capacity planning, sophisticated network design, etc. In other words, an accurate traffic prediction framework can assist ISPs with preemptive network management and ensure better network Quality of Service (QoS) and Quality of Experience (QoE) [1].

Therefore, internet traffic demand management is vital for future requirements, capacity utilization, management, planning, and optimization. Because of these reasons, research on internet traffic forecasting has gained significant interest from researchers, the ISP industry, and operational planners. However, predicting wireless internet traffic remains challenging due to the high variability and unpredictability of network traffic. Traditional forecasting methods tend to be limited in their ability to handle the complex and dynamic nature of wireless network traffic, particularly in detecting and mitigating outliers or anomalous data points.

One significant research gap in this field is the need for a multi-step wireless internet traffic forecasting model integrated with outlier point detection and mitigation techniques. Real-world internet traffic is influenced by a multitude of internal and external factors, leading to unpredictable and random characteristics [2]. Internal factors such as service launches, traffic migration, and speed upgrades, and external factors such as new internet applications, regional economic variables, and seasonal effects can result in sudden changes in traffic patterns. As a result, accurately predicting internet traffic can be challenging. Moreover, anomalies or outliers are common in real-world internet traffic [3], which can further complicate traffic forecasting. These anomalies can occur due to issues with data collection sensors, leading to faulty data being included in the analysis. To improve the generalization capability of prediction models, it is essential to identify and address anomalies/outliers in internet traffic before using any prediction model. Thus, it is critical to develop robust methods for identifying and mitigating anomalies in internet traffic to improve the accuracy of traffic forecasts. By doing so, network operators can more effectively allocate network resources, improve the quality of service for end-users, and reduce the occurrence of network failures or congestion. Outlier data points can significantly impact the accuracy of traffic forecasts and lead to sub-optimal network performance. Therefore, the development of a multi-step traffic forecasting model that incorporates outlier detection and mitigation techniques can have several significant benefits.

The proposed model should be able to accurately predict traffic patterns while identifying and mitigating outlier data points in real time. This approach will ensure that the forecasting model is robust and can provide accurate predictions, even when anomalous data points are present in the data. The use of outlier detection techniques can also provide network operators with insights into network behavior, enabling them to take proactive measures to maintain network performance. Moreover, the proposed model’s integration with outlier detection and mitigation techniques can lead to better network performance and user experience. Network operators can use the model’s predictions to optimize network resource allocation, which can lead to improved network reliability, reduced network congestion, and lower latency [4]. Therefore, the motivation for this research is to develop a multi-step wireless internet traffic forecasting model integrated with outlier point detection and mitigation techniques. The proposed model’s effectiveness will be evaluated using real-world wireless network data, and the results will be compared to traditional forecasting methods to demonstrate its superiority. This research will contribute to the advancement of wireless network technologies and provide valuable insights for network operators to improve network performance and reliability.

Wireless internet traffic forecasting is a critical task in the field of telecommunications as it enables efficient network management and resource allocation. Multi-step forecasting, which involves predicting traffic patterns for multiple time steps into the future [5], is particularly important for proactive network planning and optimization. Gradient Boosting Algorithm (GBM) is a popular machine learning technique that has shown impressive results in various forecasting tasks, including time series forecasting. Studies have demonstrated that GBM outperforms traditional forecasting methods such as ARIMA and neural networks, making it an attractive option for wireless internet traffic forecasting. However, there is a research gap in the comparative analysis of different gradient boosting algorithms for wireless multi-step internet traffic forecasting. Understanding the performance of different GBM algorithms, such as XGBoost, LightGBM, and CatBoost, can help researchers and practitioners choose the most suitable algorithm for their specific needs. Additionally, exploring the impact of different feature subsets on the performance of GBM algorithms can provide valuable insights into optimizing these models for wireless multi-step internet traffic forecasting. The research motivation for this topic, therefore, is to conduct a comparative analysis of different GBM algorithms for wireless multi-step internet traffic forecasting. The study aims to address the research gap in this area and provide insights into the most effective ways to use GBM for this task. Ultimately, this research can contribute to the development of more accurate and efficient wireless internet traffic forecasting models, which can have practical applications in telecommunications network management and resource allocation.

Our preliminary investigation [6] involved studying different machine learning models belonging to the gradient boosting and descent categories for the purpose of single-step forecasting. Additionally, we created a new machine learning model that includes an anomaly detection and mitigation module, and it was found to be superior to existing prediction models. We then proceeded to improve our anomaly detection and mitigation task by incorporating several unsupervised models and evaluating various feature sets in order to identify the ideal number of input characteristics. The following are the primary contributions of our work.

Our study involves a comparative analysis of machine learning models belonging to the gradient descent and gradient boosting categories for the purpose of predicting traffic in a single step. Additionally, we evaluate the performance of conventional machine learning models in comparison to our novel traffic prediction framework, which incorporates an anomaly detection and mitigation module.Our investigation involves the examination of actual traffic data to detect anomalous data points, utilizing both statistical and unsupervised machine learning models. Furthermore, we conduct a thorough analysis to effectively manage these outlier observations prior to incorporating them into our training models for prediction.Our study entails a thorough multi-step analysis of forecasting, wherein we assess the performance of different classical gradient descent and boosting algorithms based on their prediction accuracy and execution time. Furthermore, we evaluate the efficacy of our proposed framework across three distinct forecast lengths.Our research delves into the exploration of various feature sets for the purpose of traffic prediction. Specifically, we examine five distinct time-lagged feature subsets for predicting traffic volume in the subsequent timestamp in the context of single-step prediction. Additionally, we train our multi-step forecasting models using multiple feature sets to identify the optimal inputs that offer improved prediction accuracy and execution time.

The structure of this paper is outlined below. Section 2 provides an overview of the existing literature on traffic forecasting through the use of machine learning techniques. Section 3 details the methodology employed, encompassing the dataset used, an explanation of the machine learning models, the process of identifying anomalies, and the specifics of the experiments conducted. Section 4 explains the experimental setup for this research. Section 5 provides a summary of the performance of various machine learning models, comparing the efficacy of prediction models both with and without outliers present in the data. Section 6 concludes the paper, highlighting the direction for future investigation.

## 2. Literature Review

In this section, we discuss related works of other researchers on traffic forecasting. These works can be divided into two categories based on their prediction model type. The first category investigated different non-neural network regression and classification models for traffic prediction. The second category mostly focused on the neural network models’ implementation. We summarize the comparative analysis among existing works on traffic prediction in Table 1.

A general supervised learning technique, Gaussian Processes (GP), handles issues involving probabilistic classification and regression. Y. Xu et al. [7] presented a machine learning model based on the GP for real-world traffic prediction. The model’s performance was compared to the state-of-the-art seasonal ARIMA (SARIMA) and sinusoid superposition models. In addition, the computational complexity of identifying ideal hyper-parameters for the prediction model has been lowered from $O(n^{3})$ to $O(n^{2})$. When predicting one-hour-ahead traffic, the GP-based machine learning model displays an average prediction of 3%, which increases to 5% when the prediction period is extended to ten hours ahead. The author does not address the scalability of their proposed approach. The training and testing time of the GP-based model may increase significantly with larger datasets, making it impractical for real-world deployment. Moreover, they did not compare the proposed GP-based model with other machine learning models or state-of-the-art techniques for wireless traffic prediction. A thorough comparison could provide insights into the strengths and weaknesses of the proposed model and its potential for practical applications.

Bayati, Abdolkhalegh, and colleagues introduced a Gaussian Process Regression model (GPR) for multi-step traffic prediction [8]. Their findings were compared to classic statistical models and deep learning algorithms. The experimental findings revealed two lengths ahead of traffic prediction: 50 and 5 min. Furthermore, their proposed model provided the best forecast for normalized mean-square error (NMSE) in both situations. However, the complexity of their suggested approach is $O(n^{3})$, which means that in the event of enormous traffic data, the computational time will be significantly increased. The author does not discuss the feature selection process used in the proposed model. The selection of relevant features is critical for the accuracy of the prediction. Also, they did not justify the selected features or compare them with other feature selection methods. D. Szosta et al. [9] looked into machine learning classification and regression models for optical network traffic prediction. Four distinct regression models, linear regression, Passive Aggressive Regressor, *k* neighbors regressor, and multi-layer perceptron regressor, were used to test the proposed approach on real traffic patterns acquired from the Seattle Internet Exchange Point. Their experiment used five datasets to assess prediction performance using their proposed assessment metric, Traffic Level Prediction Quality (TLPQ). The prediction models were trained using a variety of feature sets that included window sizes and minute, day, and traffic variables. Their research claims that the regression model outperforms the categorization model in traffic prediction. D. Szostak et al. [10] reformulated the traffic prediction problem as a classification problem, and their proposed classification model predicts traffic bitrates rather than traffic volume. According to their experimental results, they utilized a Linear Discriminant Analysis (LDA) classifier to anticipate future traffic, and it was 93 percent accurate. The experiment used genuine data from Seattle’s Internet Exchange Point (which publishes its statistics online (https://www.seattleix.net/).

Different recurrent neural network (RNN) topologies have been proposed to forecast traffic flow to address the system’s temporal flexibility, such as the recurrent neural network (RNN) model. Artificial neural networks (ANNs) using recurrent networks are designed to find patterns in data sequences. These algorithms consider time sequences, incorporating both a spatial and a temporal learning dimension. Recurrent networks consider both what they have previously experienced and the current input example they are observing. Therefore, recurrent networks combine information from the recent past and the present to decide how they will react to new data. C.W. Huang et al. [11] examined the performance of three state-of-the-art deep learning models in forecasting mobile traffic statistics. The time series data’s geographical and temporal features were retrieved using the CNN and RNN models. Their study suggests a hybrid model that combines CNN and RNN and outperforms other deep and non-deep learning models. To find the best-performing model, the researchers looked at various parameter choices. Mobile devices may have limited processing power and memory, which can limit the size and complexity of the deep learning network that can be used for mobile traffic forecasting. R. Madan and P.S. Mangipudi [12] proposed an ensemble traffic prediction model combining the statistical model ARIMA and the deep learning model RNN (recurrent neural network). To separate the linear and non-linear components from the original time series data, the Discrete Wavelet Transform (DWT) transformation technique was used. The linear and non-linear components were evaluated separately using ARIMA and RNN prediction models, then combined to forecast the result. Their proposed ensemble model outperforms the individual prediction model in terms of accuracy. The proposed approach involves the use of multiple techniques, including DWT, ARIMA, and RNNs. This complexity may make it challenging to implement the approach in real-world scenarios, particularly in cases where there are limited computational resources. S. Fischer et al. [13] introduced DEEPFLOW, a traffic prediction system that uses several machine learning approaches to process ingress traffic data and anticipates all traffic flows. The prediction model is divided into two categories: statistical and deep learning models based on neural networks. They consider three statistical models as the baseline for performance comparison with different versions of sequence models, such as LSTM, Seq2SeqLSTM, and ConvLSTM. The author does not provide a detailed temporal analysis of the network traffic data, which could provide insights into patterns and trends that are not captured by the machine learning models.

An autoencoder neural network employs the backpropagation method to learn new features. The process consists of two steps: encoding and decoding. The input data are translated to a low-dimensional representation space during the encoding stage to acquire the best feature, which is then mapped back to the input space during the decoding phase. W. Wang et al. [14] introduced the SDAPM (Stacked Denoising Autoencoder Prediction Model), a unique traffic prediction model based on a Stacked Denoising Autoencoder (SDA) prediction model. From the traffic flow, the SDAPM can extract generic properties. Their model is fine-tuned using a variety of relevant hypermeters, including the number of hidden layers, the number of neurons in the hidden layer, the learning rate, and so on. The author does not discuss the impact of data preprocessing techniques on the performance of the proposed approach. Further research is needed to explore the effectiveness of different data preprocessing techniques, such as outlier detection and mitigation. T. P. Oliveira et al. [15] worked with two distinct machine learning models for traffic prediction: Multi-Layer Perceptron (MLP) and Stacked Autoencoder (SAE). They used data from a private Internet Service Provider in European towns. They employed two hidden layers of MLP with 60 and 40 neurons, respectively, and found that four hidden layers of SAE with 80, 60, 60, and 40 neurons produced the greatest results. They trained their model in both MLP and SAE for 1000 epochs, albeit the SAE training was split into two stages: unsupervised pre-training for 900 epochs and supervised training for 100 epochs. Different prediction lengths were investigated using their prediction model, and the error increased as the prediction duration increased. Their results suggest that the simpler MLP outperforms the SAE deep neural network in terms of performance. In addition, the MLP used less computational power than the SAE. The proposed approach is evaluated on synthetic traffic data and small-scale network topologies. Further research is needed to evaluate the performance of the approach on larger and more complex real-world networks. The author compares the proposed approach with only a few other methods. A more comprehensive comparison with other state-of-the-art approaches would provide a better understanding of the performance of the proposed approach. For single-step and multi-step prediction, Theodoropoulos, Theodoros et al. [16] compared the performance of the statistical model and the deep learning model. They used four distinct designs of the deep encoder–decoder model for traffic prediction. The encoder–decoder model outperformed the single-step and multi-step prediction models in the experiments. They reported a maximum of five steps ahead of traffic prediction in the case of multi-step forecasting. Overall, while the proposed approach shows promising results for predicting multi-step service traffic in a specific type of communication network, further research is needed to evaluate its performance on larger-scale networks and in comparison with other state-of-the-art approaches. Additionally, there is a need for more work to improve the interpretability of deep learning models for network traffic prediction. Our review of the existing literature on traffic forecasting tasks revealed several limitations, which led us to identify specific research gaps in this area. These gaps motivated us to propose our study to address the following areas of concern. In addition, we summarize our core contribution compared to existing works in Table 2.

While several works have been carried out in traffic prediction, most of them have overlooked the importance of outlier detection in their methodology, which can hinder the generalization of the model. The sudden changes in traffic patterns due to several factors, such as ISP companies or external events, can create outliers that affect the reliability of the model. The proposed model integrates an anomaly detection and mitigation module to address this issue, which can enhance the generalization of the model and improve the accuracy of the traffic prediction. The use of outlier detection and mitigation can help capture the underlying patterns and trends in the data and lead to more reliable and precise predictions, particularly in real-world scenarios where traffic patterns can be unpredictable and subject to sudden changes.Most existing research in the field of traffic prediction has focused on single-step forecasting and has not undertaken a comprehensive analysis of multi-step forecasting. However, multi-step forecasting is more challenging yet more practical for businesses than single-step forecasting. In our study, we addressed this limitation by considering several forecast horizons for multi-step traffic prediction tasks, in contrast to most existing works that only consider a single horizon. This approach can improve the accuracy and practicality of multi-step traffic prediction tasks, which are crucial in real-world traffic scenarios.Existing studies typically compare the performance of various machine learning models, including statistical, deep, and machine learning. However, we noticed a lack of comparative analysis between gradient boosting and gradient descent algorithms, despite their effectiveness in modeling complex data for regression problems. To address this gap, we selected several algorithms from the gradient boosting and gradient descent categories and conducted a comparative performance analysis for both single-step and multi-step traffic forecasting in our study. This approach can help identify the most efficient and accurate algorithms for traffic prediction tasks, especially in scenarios where data complexity is a challenge.Unlike most existing studies that rely on synthetic traffic datasets for their analysis, we utilized a real-world traffic dataset in our experiment. This approach can enhance the practicality and relevance of our findings and provide a more accurate representation of real-world traffic scenarios.
sensors-24-01871-t001_Table 1Table 1Summary of internet traffic forecasting techniques: a comparative analysis.Ref.MethodologyData PreprocessingForecast
HorizonDatasetOutlier
DetectionEvaluation Metric[7]Gaussian processesBased stations are grouped based on geographical locationSingle-Step and multi-stepUnivariate, temporalN/AMean Absolute Percentage Error (MAPE)[9]linear regression, Passive Aggressive Regressor, k neighbors regressor, and multi-layer perceptron regressorN/ASingle-StepUnivariate, temporalN/ATLPQ (Traffic Level Prediction
Quality), Root Mean Square Percentage Error (RMSPE)[11]RNN and CNNN/ASingle-StepMulti-variate, temporal, and spatialN/AMAE (Mean Absolute Error), RMSE (Root Mean Square Error), MAPE, and MA (Mean Accuracy)[12]ARIMA and RNNDiscrete wavelet transform DWT) decomposes a given discrete signal into orthogonal wavelet functions.Single-StepUnivariate, temporalN/ANRMSE (Normalized Root Mean Square Error)[13]LSTM, Seq2SeqLSTM, ConvLSTMreordering of flow streams, window-based value aggregation, and harmonizationMulti-stepMulti-variate, temporalN/ARMSPE[14]Stacked Denoising Autoencoders and MLPMin-Max Normalization, handling missing valueSingle-StepUnivariate, temporalN/AMAE, RMSE, MRE (Mean Relative Error)[17]echo state networkPhase-space reconstruction has been used to reconstruct the original network traffic data seriesMulti-stepUnivariate, temporalN/ARMSE, MAE[18]LA-Resnetmissing data were replaced and filled to the average of nearby values.Single-stepUnivariate, temporal, spatialN/ARMSE, MA[19]LSTM and Online-Sequential Extreme Learning Machine (OS-ELM)Data transformation for supervised learning, data differencing, and normalizationSingle-stepUnivariate, temporalN/AMSE[20]TTGCN (Temporal–Topological Graph Convolutional Neural Network),N/AMulti-stepUnivariate, temporal, spatialN/AMAE, RMSE[21]LSTM encoder–decoderMin-Max NormalizationMulti-stepUnivariate, temporalN/ARMSE, R^2^ (R Squared)[22]Linear Regression, KNN, and Random ForestSeasonality feature extractionSingle-StepUnivariateN/ARMSPE
sensors-24-01871-t002_Table 2Table 2Comparison of core contributions against existing works.Key ContributionExisting WorksProposed WorkOutlier Detection and MitigationLacks comprehensive methods for identifying and mitigating outliers, potentially leading to inaccurate traffic forecasting.Employs empirical analysis and unsupervised learning for robust outlier management, enhancing forecasting accuracy.Forecast HorizonPrimarily focuses on short-term predictions, with minimal exploration of long-term forecasting challenges.Includes both short-term and long-term forecasts, examining the impact of forecast horizon on accuracy.Feature OptimizationLimited investigation into the optimal selection of features for improving model performance.Conducts experiments with various feature subsets to identify the most effective inputs for forecasting.Model SelectionComparisons often span across broad categories, lacking depth within specific model types for traffic forecasting.Provides a detailed analysis within the boosting model category, offering insights into achieving superior prediction accuracy.


## 3. Methodology

In this section, we describe our proposed traffic prediction framework, depicted in Figure 1. Firstly, we discuss our data preprocessing steps in Section 3.1 and Section 3.2. Then, we describe the anomaly detection and mitigation techniques to manage outlier points in our traffic data in Section 3.3. Next, the data windowing or feature extraction module is explained in Section 3.4. Finally, we explain the traffic prediction module of our model in Section 3.5.

### 3.1. Handling Missing Value

In our experiment, we eliminated the final day’s data from the dataset because the network’s trace was only partially obtained on that day. In our sample, twenty-nine missing values are filled in using the forward filling method. The missing number in our traffic data has been replaced with the most recent instance of valid data. However, other techniques exist to deal with missing data in time series analysis, including linear interpolation, quadratic interpolation, replacing with mean value, etc. The linear interpolation method calculates a null value by linking data points in a straight line while assuming that there is a linear correlation between them. But our non-linear dataset led us to conclude that this approach was ineffective for treating missing values. Additionally, it appears that the polynomial interpolation method is inappropriate since, in order to utilize this method to replace the missing data, we must first provide the order. The smallest probable degree that goes through the available data points is used to fill in the missing numbers. Since there may be outliers or other unusual data points in the real-world traffic measurements, we do not consider the mean value to alter the missing numbers. We discovered that the forward filling technique [23], which is commonly applied to time series data, was beneficial for our experiment. This approach assumes that the traffic volume would have remained stable from the point of dropout until the end of data collection, and it fills in any missing values after the dropout with the most recent measurement. It also relies on the assumption that missing data are completely random, meaning that the likelihood of missing values is not influenced by factors that impact traffic volume at a specific time.

### 3.2. Autocorrelation Function (ACF) Analysis

The ACF is widely used to assess the data in time series analysis and prediction. The ACF plot visually displays the degree of correlation between an observation in a time series and observations made at earlier time steps. The underlying time series must be stationary for ACF to operate. The ACF plot identifies the correlation between a time series data point and previous data points, called lag (*l*), of the same time series. For a given set of lags, the ACF evaluates the correlations among samples in a time series. The ACF for time series *x* is given by:(1)$$ACF=Corr(x_{t},x_{t-l}),\;\mathrm{where}\;l=1,2,3\ldots$$

We can determine the stationarity and randomness of time series data using an ACF plot. Also, the time series seasonality and trend can be identified based on ACF plot information. Each bar in an ACF plot indicates the strength and direction of the correlation among data points. The bar value should be near zero for all lags for random data, which we can also consider as white noise. Non-random time series data should have at least one lag value with a strong correlation. We can use time-lagged features in building a prediction model for non-random time series data. Our traffic data have many lags with strong correlations, which indicates that the data are non-random. Therefore, we considered previous timestamp features for our regression models to predict the following values. Based on our ACF plot, we now analyze two important time series characteristics: seasonality and trend. Smaller lags frequently exhibit strong associations when trends are prevalent in the time series because samples closer in the period tend to have comparable values. As the lags lengthen, the correlations gradually diminish. When periodic patterns are evident, multiples of the frequent recurrence have stronger autocorrelations than other lags. The ACF plot blends both characteristics when a time series contains a trend and seasonality. We can conclude that our traffic data have trend and seasonality based on the correlation value for different lags. Since the correlation is higher for smaller lag and decreases for larger lag, there is a trend in our traffic. Also, there is a repetition of higher correlation values for every 288 lag values for our traffic dataset, in which samples are collected every five-minute interval. In other words, we find a daily periodic pattern in our traffic data. So, ACF plot analysis gives us several important pieces of information about our traffic data, such as the fact that our traffic data are non-random and have a trend and daily seasonality. Also, it helps to decide that we can better model our regressor model based on the time-lagged feature. Hence, we investigated several lagged feature sets for single and multi-step forecasting models for optimum input settings.

### 3.3. Anomaly Detection and Mitigation

In this subsection, we discuss our methods to detect those data points that deviate from most of the data instances in the dataset. Those unexpected data points are called anomalies or outliers. Many external and internal factors make real-world IP internet traffic susceptible. These issues disrupt normal traffic flow, which must be discovered and corrected so that a machine learning model may improve its generalization capability. There are different types of anomalies: point anomalies, contextual anomalies, and collective anomalies. In our research, we considered only the point anomalies, i.e., those data points which are far away from the general distribution of the data. There are mainly three main categories of anomaly detection methods: statistical profiling, supervised learning, and unsupervised modeling.

In statistics, the three-sigma rule is a statistical calculation that defines the upper limit and lower limit for point anomaly detection based on three standard deviations from the mean value of the dataset. The data points outside the boundary defined by the three-sigma rule are considered outliers in the dataset. Therefore, it is necessary to calculate the absolute difference for each data point and their average; if the difference is within three standard deviations of the dataset, it is considered a statistically valid data point. According to the three-sigma rule, the probability of a data point *X* lying within three standard deviations of the mean is 99.73%. In our experiment, we applied this empirical method for identifying point anomalies from our dataset before providing them to our machine learning models. Using the three-sigma rule assumes that the probability distribution of the data follows a normal distribution. It is true that data that follow a normal distribution respond to the three-sigma rule the best. However, even with non-normally distributed variables, the Bienaymé–Chebyshev inequality, sometimes known as Chebyshev’s inequality, states that at least 88.8% of cases should fall inside correctly computed three-sigma ranges. For a wide range of distinct probabilities, Chebyshev’s inequality states that a minimum of just 75 percent of observations must reside within two standard deviations of the mean and 88.89 percent within three standard deviations [24,25]. We applied three different unsupervised machine learning models from the clustering category, although there are other categories: model-based, graphical, distance-based, and supervised learning. Isolation Forest (ISoF), K-Nearest Neighbors (KNN), and Clustering-Based Local Outliers (CBLO) are selected for our experiment. The clustering-based approach grouped all data points into several clusters, and the data instances that do not belong to these clusters are called outliers. Generally, it is challenging to define and identify the outliers from the datasets, and that is why we compared the performance of different models to evaluate their corresponding results. For the sake of the analysis, the contamination percentage in the data is set at 1%. After studying the associated temporal information, we discovered that the data points recommended by the three-sigma rule are more likely to constitute the anomaly.

### 3.4. Feature Extraction

For supervised learning applications, it is essential to format time series data correctly. Typically, time series data are organized as a sequence of tuples (time, value), which is not directly suitable for use in machine learning algorithms. To address this, we adapted our time series data using the sliding-window approach, a method depicted in Figure 2. In this approach, each instance of time series data is indicated by $t_{i}$, with *i* representing the data’s index. For instance, as shown in Figure 2a, we treat the initial three data points as a feature set $X_{1}$ to forecast the fourth data point, which is marked as the target $y_{1}$. The term “window width” or “lag size” refers to the count of previous time steps utilized to predict the subsequent time step. This procedure is repeated, advancing one step at a time, until the final data point in our training dataset is used for prediction. This demonstrates the application of the sliding-window method for single-step forecasting. For multi-step forecasting, the target comprises multiple future values. We have prepared our data for both single-step and multi-step forecasting, as further detailed in Section 3.4.1 and Section 3.4.2.

#### 3.4.1. Feature Extraction for Single-Step Prediction

We consider different window widths for single-step prediction to predict our next step. We performed a grid search to find out the best window width for single-step data conversion. A set of five different window widths as {6, 9, 12, 15, 18} is considered for our experiment, which indicates five different data conversion configurations for single-step prediction. The initial data configuration consists of six features to predict the target, and it continues to eighteen features for identifying the optimized window width for the sliding technique.

#### 3.4.2. Feature Extraction for Multi-Step Prediction

In the case of multi-step prediction, previous time steps are used to predict the next two or more steps, known as the forecast horizon. Figure 2b illustrates the process of feature extraction for multi-step forecasting. We considered three different time steps in our experiment: six-step-, nine-step-, and twelve-step-ahead forecast. In addition, we explored various feature sets based on forecast length to find optimum inputs for the model. For example, we searched through a window width set of {6, 9, 12, 15, 18} for six-step prediction to identify the optimum number of inputs for six-step forecast models. Therefore, our proposed prediction model’s performance has been evaluated based on multiple combinations of (features and targets). We provided five different varieties of feature and target variables, including {(6, 6), (9, 6), (12, 6), (15, 6), and (18, 6)}, to our prediction model to find the best input set for six-step prediction. The exact process has been followed for other forecasting lengths in our experiment. For example, a total of four and three different (features, input) combinations have been considered for nine- and twelve-step forecasting models. We analyze the performance of each combination for every particular forecast horizon and identify the best input settings for the corresponding model.

### 3.5. Regression Models for Single- and Multi-Step Forecasting

In predictive modelling, both single- and multi-step forecasts can be proposed within the context of regression problems, each with its unique set of objectives and methodologies.

For single-step prediction, the aim is to predict a single future value based on past input features. This involves learning a function *f* that maps a set of input variables *X* to a single output variable *y*. The relationship can be mathematically expressed as:y=f(X)

The goal is to accurately predict the next value in the sequence, optimizing the prediction accuracy as evaluated by metrics such as Mean Squared Error (MSE), Root Mean Squared Error (RMSE), or Mean Absolute Percentage Error (MAPE). Extending this concept to multi-step predictions, the task evolves into a multi-output regression problem. Here, the dataset consists of input variables *X* and output variables *Y*, with *Y* being a matrix with *m* columns, corresponding to *m* future time steps to be predicted. The objective becomes to learn a function *f* such that:$$\hat{Y}=f\left(X\right)$$
is a close approximation of *Y*, where $\hat{Y}$ represents the structure of *Y* with the same number of columns. Each column in *Y* and $\hat{Y}$ represents a future time step, making $\hat{Y}$ a comprehensive prediction across multiple future points. This can be represented as:(2)$$Y=[y_{1},y_{2},\ldots ,y_{m}]$$
(3)$$\hat{Y}=\left[\hat{y_{1}},\hat{y_{2}},\ldots ,\hat{y_{m}}\right]$$
with each $y_{i}$ and $\hat{y_{i}}$ denoting the actual and predicted values for the *i*th future time step, respectively. The challenge is to minimize the gap between *Y* and $\hat{Y}$, employing loss functions similar to those used in single-step prediction to measure and optimize the accuracy of predictions. Through this framework, both single- and multi-step predictions aim to employ past data to forecast future values.

In our single-step prediction approach, we employed a group of five various regression models, each with its unique strengths and methodologies. The Gradient Boosting Regressor (GBR) leverages gradient boosting to optimize arbitrary differentiable loss functions, building an additive model progressively while fitting a regression tree at each stage to the negative gradient of the loss function. Similarly, Extreme Gradient Boosting (XGB) offers a robust implementation of gradient boosting designed for both classification and regression tasks, noted for its computational efficiency relative to other gradient boosting frameworks. Light Gradient Boosting Machine (LGB) addresses the constraints of histogram-based models by integrating innovative techniques such as Gradient-based One Side Sampling and Exclusive Feature Bundling, enhancing its performance. CatBoost Regressor (CBR), a binary-tree-based gradient boosting model, tackles common gradient boosting challenges by introducing an ordering principle, ensuring reliability in predictions based on the sequencing of training sample targets. Lastly, Stochastic Gradient Descent (SGD) presents a straightforward yet powerful approach to learning, differentiating itself by randomly selecting data samples for gradient computation in each iteration, thereby optimizing computational efficiency.

In extending our analysis to include multi-step forecasting, we adopted a multi-output regression framework. This approach is compatible with several machine learning models, including linear regression, decision tree regressor, and random forest regressor, which are capable of predicting multiple outputs simultaneously. However, not every regression model, such as support vector regression, inherently supports multi-output predictions. To address this limitation, these models can be adapted through various methods. A common strategy involves segmenting the multi-output regression challenge into several single-output regression problems. This method, often referred to as direct multi-output regression, treats each output as independent and predicts each step using the same set of input data. There is an extension of this approach where individual model outputs are connected to each other. A sequence of regressors can be employed to solve multi-output regression problems, where each regressor in the sequence learns the relationship between the input variables and a specific output variable. The first regressor learns the relation between the inputs and the first output, and the subsequent regressors use the inputs and the outputs predicted by the previous regressors to learn the remaining output variables. The final regression model then uses all the input variables and the predicted outputs from the previous regressors to predict the final output. This approach is known as chained multi-output regression [26], as depicted in Figure 3. We applied the second strategy for our multi-step prediction problem, as there is a correlation between traffic volume for two consecutive timestamps. However, one major drawback of this approach is that the order in which the output variables are arranged in the sequence can have a significant impact on the accuracy of the predicted outputs.

### 3.6. Proposed Model’s Algorithmic Analysis

As defined in Algorithm 1, our proposed traffic prediction model takes traffic data, denoted as $y_{t}$, as input for feature extraction, model training, and validation. After performing a comprehensive analysis, it returns the best-performing model based on prediction accuracy and execution time. Our algorithm begins by handling missing values in the traffic data, as shown in the code block in lines 5 to 9. Then, in lines 10 to 12, we loop through each anomaly detection algorithm from the unsupervised category to find outlier data points from the traffic data. Additionally, the statistical profiling technique, called the three-sigma rule, has been applied to identify outlier data points based on lower and upper bounds, as described between lines 13 and 21. Finally, we convert our traffic data into time-lagged features and targets to apply a supervised forecasting model to predict different forecast horizons. Code blocks from lines 22 to 33 represent the feature extraction, model development, and model evaluation part of our algorithm.
**Algorithm** **1:** Traffic prediction integrated with anomaly detection and mitigation.
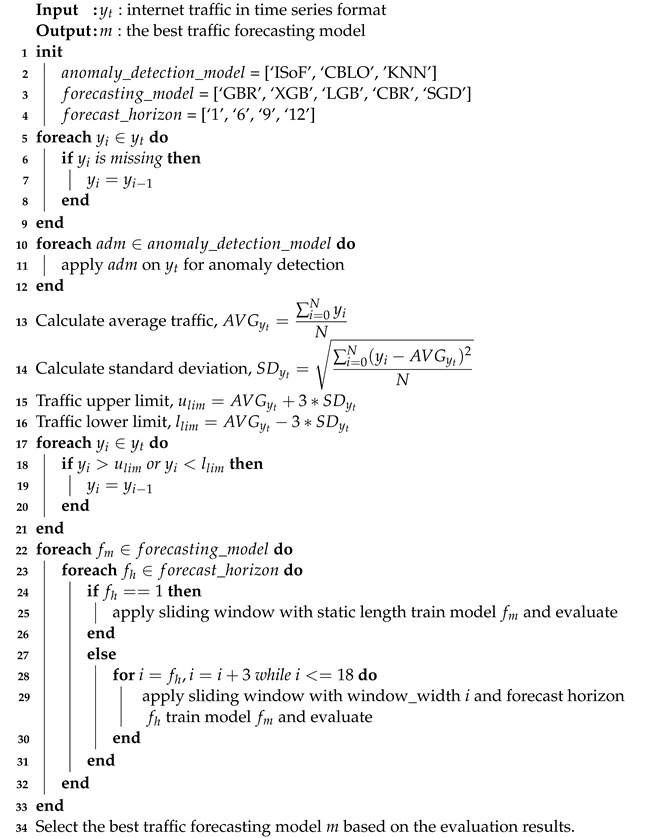


The total time complexity of our proposed model is the summation of the time complexity of handling missing values, anomaly detection, anomaly mitigation, feature extraction, and machine learning algorithm. Missing value imputation and anomaly mitigation take linear time to complete the operation, i.e., their worst-case time complexity is O(n). The time complexity of the anomaly detection algorithm depends on several factors. For example, in the case of KNN and CBLO, the number of clusters, *k*, impacts the running time of the algorithm. The worst-case time complexity of KNN consists of two individual operation costs: nearest neighbors model fitting and the distances of each point to its *k*th nearest neighbor. Therefore, the overall KNN worst-case time complexity is O(nlogn+nlogk). Similarly, the time complexity of the CBLO algorithm also depends on *k*, and the worst-case time complexity is $O(n^{k+1})$. However, in our model, isolation forest takes the maximum worst-case time complexity, which is $O(n^{2}logn)$. The time complexity of the feature selection module depends on the window size, *w*, as we used the sliding-window technique to extract time-lagged features from our traffic dataset. Our proposed model takes O(n∗w) time to extract features with a window width of *w*. Finally, the time complexity of our machine learning model depends on the number of total trees, the depth of each tree, the number of features, and the total number of samples. Moreover, the training and prediction time complexity may vary based on the specific machine learning algorithm used. Therefore, we consider the machine learning algorithm with the highest worst-case time complexity, denoted as $O(f_{m})$, among all the forecasting models. The overall time complexity of our proposed model is summarized below.
(4)$$\begin{array}{cc} T(n) & =O\left(n\right)+O\left(n^{2}logn\right)+O\left(n\right)+O\left(n\times w\right)+O\left(f_{m}\right) \end{array}$$

## 4. Experimental Detail

In this section, we provide an overview of our experimental setup. We begin by describing our dataset in Section 4.1. Then, we explain the cross-validation technique used to validate the model’s performance in Section 4.2. Finally, we summarize the performance evaluation metrics for single-step and multi-step forecasting in Section 4.3 and provide details on the required software and hardware configurations in Section 4.4, respectively.

### 4.1. Dataset Description

For this experiment, real internet traffic telemetry on a number of high-speed ports was employed. The value of the ifOutOctets counter of a core-facing interface on a provider edge router’s SNMP (Simple Network Management Protocol) interface was sampled in order to gather telemetry data. The bps (bit per second) value for the interval is the gap between the observations at either end of the interval multiplied by 8. Samples are collected every five minutes. There were never any skips during the sampling time because this was a 40 Gbs connection. Our dataset contains 8563 data samples, with 29 days of complete data (288 data instances each day) and 1 day of missing information. Only the timing (GMT) and traffic information from the actual data source in JSON (JavaScript Object Notation) format were taken into account; all other data were ignored. In the end, 29 days’ worth of data were taken into consideration for constructing our prediction model. In this section, we examined our data to identify any missing values and took the appropriate steps to fill them in.

### 4.2. Time Series Cross-Validation

Our experiment used a rolling-based cross-validation method to assess the accuracy of our traffic forecast models, as depicted in Figure 4. To train and evaluate classification/prediction models, there are numerous approaches to dividing the dataset into multiple folds. *K*-fold cross-validation divides the dataset into *K* folds of about equal size, with all except one used to train the model and the remaining fold applied for testing. The procedure is iterated until the entire model has been tested on each fold, and the model’s ultimate result is calculated as the average results on each fold.

Because most machine learning cross-validation algorithms choose folds at random, we need to take a different strategy for dividing and picking folds from time series data in order to preserve the temporal relationship between them. We apply a rolling basis cross-validation method in which training begins by one fold and ends with the prediction of the following fold. The previous stage’s test fold is incorporated into the training process, and the subsequent fold for testing in the following step. The model’s final performance is the mean of each fold’s prediction.

### 4.3. Evaluation Metrics

The performance of our traffic forecasting models was estimated using Mean Absolute Percentage Error (MAPE). The performance metric identifies the variation in the anticipated result from the original data. MAPE, for example, is the average percentage of the variance between the actual and predicted values. As a result, we can formally define our performance metric MAPE and Mean Accuracy (MA) for single-step forecasting, as follows, where $p_{i}$ and $o_{i}$ are the predicted and original values, respectively, and *n* is the total number of test instances:(5)$$MAPE=\frac{1}{n}\sum_{i=1}^{n}|\frac{p_{i}-o_{i}}{o_{i}}|\times 100$$
(6)MA=(100−MAPE)%

The MAPE formula for multi-step prediction is similar to the formula for single-step prediction, but it considers all the predicted values over the forecast horizon. Assuming actual traffic volume values $o_{t}$ and predicted values $p_{t+h}$ for *h* steps ahead, the formula for $MAPE_{h}$ for multi-step prediction is:(7)$$MAPE_{h}=\frac{1}{n}\sum_{t=1}^{n}\left|\frac{o_{t}-{\hat{p}}_{t+h}}{o_{t}}\right|\times 100$$

We can define $MA_{h}$ for multi-step forecasting as the average $MA_{i}$ of each individual step prediction, *i*, in forecast horizon *h*.
(8)$$MAPE_{avg}=\frac{1}{h}\sum_{i=1}^{h}MAPE_{i}$$
(9)$$MA_{h}=\left(100-MAPE_{avg}\right)\%$$

We also consider the model execution time for measuring their comparative performance.

### 4.4. Software and Hardware Preliminaries

Our experiments were carried out using the Python programming language (version 3.9.13) and the scikit-learn machine learning library (version 1.3.2) [27]. The tests were run on a computer equipped with an Intel (Santa Clara, CA, USA) (R) i3-8130U CPU at 2.20 GHz, 8 GB of RAM, and a 64-bit Microsoft (Redmond, WA, USA) Windows operating system.

## 5. Result and Discussion

The first 21 days of IP traffic, that is, 70% of our total data, were utilized for training our machine learning models, while the remaining 30% of the data, i.e., eight days, were used for testing. Our investigation is divided into two phases: (I) machine learning model performance evaluation for single-step traffic prediction, and (II) machine learning model performance evaluation for multi-step traffic prediction, as described in subsections in Section 5.2 and Section 5.3, respectively. Before that, we discussed the impact of outlier mitigation on data variability in Section 5.1.

### 5.1. Outlier Mitigation Impact Analysis

The outlier replacement on our dataset, resulting in a change in the mean and standard deviation of the data, is shown in Table 3. In addition, we depict the actual traffic and outlier mitigated traffic to show their variability in Figure 5. To assess whether this change is statistically significant, we conducted three hypothesis tests: a two-sample *t*-test, a Wilcoxon rank-sum test, and a Kolmogorov–Smirnov test, and the results are summarized in Table 3. Before performing these tests, we compared the distribution of our dataset before and after outlier treatment, as depicted in Figure 6.

The ECDF plot captures a clear visual difference in the range of the two datasets before and after outlier replacement. The ECDF plot suggests that the range of values is narrower after outlier replacement, indicating a reduction in variability. While the removal of the outlier reduced the range of values, it may not have had a substantial impact on the overall variability of the data. To determine this fact, we performed three different statistical tests to examine the overall shape of the distributions, rather than just the range of values. It is possible for the standard deviation of the two groups to be similar, but for the range of values to differ due to the presence or absence of an outlier.

The two-sample *t*-test is used to compare the means of two independent samples. In this case, the two samples are the data before outlier replacement and the data after outlier replacement. The test statistic is the t-statistic, which measures the difference between the means relative to the variability within the samples. The *p*-value is the probability of obtaining a test statistic as extreme as the one observed, assuming that the null hypothesis (no difference in means) is true. The *p*-value for the two-sample *t*-test is 0.7425, which is greater than the conventional threshold of 0.05 for statistical significance. This suggests that there is insufficient evidence to reject the null hypothesis of no difference in means, and that the difference between the means is likely due to chance. The Wilcoxon rank-sum test, also known as the Mann–Whitney U test, is a non-parametric test that compares the medians of two independent samples. Like the *t*-test, it can be used to test for a difference in location (i.e., central tendency) between two groups. The *p*-value for the Wilcoxon rank-sum test is 0.9877, which is much greater than 0.05. This also suggests that there is insufficient evidence to reject the null hypothesis of no difference in medians. The Kolmogorov–Smirnov test is a non-parametric test that compares the distributions of two samples. It is based on the maximum difference between the cumulative distribution functions of the two samples. The *p*-value for the Kolmogorov–Smirnov test is 0.9999, which is very close to 1. This also suggests that there is insufficient evidence to reject the null hypothesis of no difference in distributions. In summary, based on the results of these three tests, there is no significant evidence to suggest that the outlier replacement has had a significant effect on the central tendency or distribution of the data. Based on the statistical tests we performed, it appears that the variability of our traffic data has decreased after the outlier replacement. However, the difference in standard deviation is not statistically significant, as indicated by the *p*-values of the tests. The fact that these tests did not find a statistically significant difference in the standard deviation of the two groups suggests that the reduction in variability after outlier replacement is not likely to be due to chance. Therefore, it may be appropriate to interpret the results as follows:The ECDF plot suggests that the range of values is narrower after outlier replacement, indicating a reduction in variability.The statistical tests indicate that there is no significant difference in the standard deviation of the two groups, suggesting that the difference in range is not likely to be due to chance.Taken together, these results suggest that the removal of the outliers has resulted in a reduction in the range of values, even though the standard deviation of the two groups is similar.

Based on the results, it appears that outlier replacement has some impact on reducing the variability of the data, but the magnitude of this reduction may not be large enough to be statistically significant.

### 5.2. Single-Step Traffic Prediction

To conduct our traffic prediction experiment, we divided it into two parts: single-step and multi-step forecasting. The single-step model predicts the next time step based on historical data, while the multi-step model forecasts more than one step at a time. Our multi-step forecasting experiment considered different forecast lengths of six, nine, and twelve steps. We evaluated and compared the performance of several models from the gradient boosting category, such as Gradient Boosting Regressor (GBR), XGBoost (XGB), LightGBM (LGB), CatBoost (CBR), and the gradient descent category, such as Stochastic Gradient Descent (SGD), for both single-step and multi-step forecasting. We also proposed a machine learning architecture by integrating an outlier detection module with these models and compared their performance with state-of-the-art gradient boosting and descent methods. Moreover, we experimented with various feature subsets to train our models to identify the optimal input for better prediction.

Table 4 compares the performance of the standard machine learning model with our proposed model for single-step prediction. Table 5, Table 6 and Table 7 summarize the performance for six-step, nine-step, and twelve-step forecasting, respectively. We report the mean absolute percentage error between actual and predicted traffic for each step in multi-step forecasting, for different input lengths. The best performance for the corresponding model is highlighted in bold and underlined. Our single-step traffic prediction experiment began with the GBR model trained using five different feature subsets of time-lagged historical data points. The input column represents the number of previous time-dependent values used to predict the next value. For example, an input value of six means six past observations were used to train our model to predict the following data. We used mean absolute percentage error to measure the model performance, which represents the average difference between actual and expected observations.

Firstly, we analyze and discuss our prediction model performance in single-step forecasting. The minimum average gap between actual and predicted traffic using the standard GBR model is 7.47% for input length six. Compared to our proposed GBR model, where we identify and mitigate outlier samples before using them to train our model, we achieved 30% less error of 5.20% using similar input settings. We expand this experiment with a different boosting algorithm to compare prediction accuracy. XGB is one of the popular implementations of gradient boosting to solve the inefficiency of creating a new branch in their tree-based algorithm. Our proposed XGB model gave better prediction accuracy than GBR for every corresponding input setting. The lowest error for XGB is 5.15% after handling the outlier data points, while the standard best-performing model prediction error is 7.47%. Therefore, our proposed XGB model with anomaly detection module gave 2.3% more prediction accuracy than XGB. But the best performance of standard XGB and GBR is identical with a minimum gap of 7.47%, but XGB performs a little better in almost every input configuration except six. Hence, we can conclude that our proposed XGB outperforms standard XGB and GBR, while the classical XGB performance is very close to classical GBR based on prediction accuracy. However, we also consider their execution time with the prediction accuracy in multi-step forecasting for comparing their performance. Then, we consider another variation of gradient boosting called LGB. Our proposed LGB performs better than both GBR and XGB for different feature sets. The best LGB prediction accuracy is 94.92% with a minimum average gap of 5.09%. Both GBR and XGB split the tree level-wise, while LGB follows the leaf-wise splits, and reducing more loss results in better performance. Finally, we consider CBR, another version of gradient-boosted tree algorithms, for our single-step traffic prediction. According to the original work [28], the CBR was proposed for categorical data processing and was shown to outperform XGB and LGB using eight datasets. Since CBR is also used for feature ranking [29] and classification purposes [30], we considered it for our traffic forecasting task. According to our experiment, CBR is the best-performing model among all considering approaches from the gradient boosting category, with a minimum average gap of 5.08%. Compared to the standard CBR model performance, our proposed CBR provides around 33% less prediction error. Our experimental results reveal an overall performance gain for all gradient boosting models integrated with the anomaly detection and mitigation module. Based on our findings, we can conclude that removing outliers from the dataset before training the prediction model can significantly enhance the prediction accuracy, by approximately 30% on average. We also compare the performance of different gradient boosting regressor models with the SGD model. As SGD follows a fixed model architecture, its average prediction accuracy is lower by more than 3–4% compared to gradient boosting algorithms. But our proposed SGD’s best performance improved significantly in terms of error, from 10.44% to 6.10%, and reduced the accuracy gap to approximately 1% compared to GBR and its variants.

### 5.3. Multi-Step Traffic Prediction

Now, we analyze the detailed prediction results for multi-step forecasting. In the case of six-step forecasting, our proposed GBR model performs better for each step. The minimum prediction error we achieved was with eighteen input data points, where the last prediction accuracy was around 93%, with an average error of 7.03%. The individual step prediction error increases with larger forecast steps as the prediction error in the current steps accumulates on the next step prediction. As a result, multi-step forecasting is more challenging than single-step forecasting. In the case of nine- and twelve-step forecasting, our proposed GBR model performs better, with a minimum average gap of 8.88% and 10.60%, respectively, between actual and predicted traffic for the last step. Next, we investigated the performance of the XGB model and noticed a general performance improvement compared to GBR. For example, in the case of six-step forecasting, our proposed XGB model provided the lowest average gap of 6.99% with eighteen features, which is better than the GBR model’s performance and standard XGB performance. XGB and GBR both follow the same principle of gradient boosting. But XGB can control over-fitting by formalizing a more regularized model, which results in better prediction accuracy. As a result, we noticed an outperformance of the XGB model over GBR. Compared to nine- and twelve-step forecasting, our proposed XGB performs better than GBR for twelve-step prediction only. But GBR took approximately 1.5 times more execution time than XGB for nine-step forecasting, as depicted in Figure 7. Moreover, in terms of the execution time for the six- and twelve-step forecasting models, GBR execution time is 1.8 and 1.7 times greater than XGB. XGB takes less time for execution than GBR because XGB reduces the size of the search space for possible splitting by considering the distribution of the features across all data samples in the tree leaf. Overall, XGB’s performance is better than GBR when evaluating prediction accuracy and execution time.

Now, we analyze the performance of another model in the gradient boosting category, called LightGBM (LGB). Compared with XGB and GBR prediction performance, our proposed LGB gave the minimum average prediction error of 6.30% and 7.18% for nine- and twelve-step prediction, respectively. In contrast, for nine and twelve steps, the best performing GBR and XGB prediction errors are 6.33% and 7.26%, and 6.32% and 7.25%, respectively. For six-step prediction, the LGB average prediction accuracy was 94.64%, the same as GBR, while the XGB provided the highest average accuracy of 94.65%. Overall, LGB has better prediction accuracy compared to GBR and XGB. LGB performs leaf-wise growth as opposed to level-wise expansion in XGB, which produces larger loss reduction and much more complex trees and, as a result, improved accuracy, while also being faster. Also, we noticed a slow error propagation rate between consecutive forecast steps in LGB compared to GBR and XGB. For example, in nine-step forecasting, our proposed XGB model started with a better first step prediction, with an error of 3.54%, which is less than the LGB first step error of 3.55%, but for the last step, the XGB error was 8.90%, which is higher than LGB’s last step error of 8.72%. According to Figure 3, LGB execution time is less than GBR and XGB. After analyzing the execution time for each forecast length, we concluded that LGB execution time is a minimum of 2 and 1.5 times shorter than GBR and XGB. LGB is a histogram-based approach that accelerates the training process by bucketing continuous attribute values into discrete bins. Finally, our last model from gradient boosting is CatBoost (CBR). CBR is the best-performing model with and without outlier detection among all considered models in our single-step and multi-step forecasting experiment. The best average prediction accuracy using our proposed CBR is 94.70%, 93.78%, and 92.89%, respectively, for six-, nine-, and twelve-step forecasting. All best-performing CBR models provided the highest accuracy with fifteen features. A weighted sample variation of SGD, known as minimal variance sampling (MVS), is provided by CBR. This method uses weighted sampling at the tree layer rather than the split level. To increase the precision of split grading, the data for each boosting tree are chosen in a certain way, resulting in better accuracy. However, CBR takes the highest execution time among all prediction models. For example, LGB is the fastest algorithm from the boosting category, and CBR execution time is more than three, four, and six times more than LGB, respectively, for six-, nine-, and twelve-step forecasting.

Lastly, we consider a simple machine learning model called stochastic gradient descent (SGD) for our traffic prediction task. Since gradient boosting and gradient descent both work similarly and descend the slope of the loss function, we consider SGD for our experiment to show a comparative analysis with the gradient boosting algorithm. According to our investigation, the SGD prediction is the lowest among all prediction models for single-step and multi-step forecasting. For example, the best standard SGD model performance for twelve-step forecasting is 85.64% without outlier detection, which is more than 6% less than the gradient boosting algorithm. But our proposed SGD model performance improved significantly compared to classical SGD, with an average accuracy of 91.90% for twelve-step prediction, but still, it is smaller than gradient boosting’s best performance. Gradient boosts the descent gradient by adding new models, as opposed to gradient descent, which descends the gradient by introducing modifications to hyper-parameters. As a result, the model architecture gradient boosting changed dynamically, as opposed to the fixed model in SGD, which gave us better prediction using gradient boosting than SGD. However, the SGD is the fastest algorithm among all prediction models from the boosting category.

We also investigated the average accuracy for each input set. Then, we calculated all individual step prediction accuracies to determine the average accuracy for particular input and model settings. Figure 8, Figure 9 and Figure 10 illustrate the average accuracy comparison between the standard model and our proposed model for each feature set. From these figures, we noticed that our proposed model outperforms traditional models for each input configuration, and the average accuracy increases with the larger feature set, except with the CBR model. In the case of the CBR model, the model performance dropped for eighteen time-lagged features, which is common for each multi-step forecast window. However, SGD showed a completely different behavior compared to gradient boosting algorithms. Furthermore, the standard SGD model’s average prediction accuracy decreases with an increase in the input size, while our proposed SGD model’s average accuracy increases with the input size. Therefore, we can conclude that the anomaly mitigation module helps the model to netter learn the traffic pattern, resulting in better accuracy.

In the pursuit of enhancing the performance of our multi-step forecasting models, we undertook a comprehensive evaluation and comparative analysis of several different prediction models, focusing on their accuracy and execution time in the presence and absence of outliers. The results from this analysis not only highlight the strengths and shortcomings of these models, but also underline crucial observations that could guide future research in this field. In the following discussion, we summarize these key insights and their implications for further exploration and refinement of multi-step forecasting models.

Improved Accuracy with Outlier Removal: The results highlighted that models trained without outlier data generally performed better than their counterparts trained with outliers. This emphasizes the importance of data preprocessing and outlier removal in enhancing model performance.Prediction Challenges in Multi-Step Forecasting: The study pointed out the increased complexity and challenges associated with multi-step forecasting as compared to single-step forecasting. This is due to the accumulative error in the current steps, which impacts the next step prediction.Impact of Feature Selection: The study found that the number of input features used can significantly influence model performance. This signifies the importance of feature selection and dimensionality reduction techniques in predictive modeling.Trade-off between Accuracy and Execution Time: Different models exhibited trade-offs between prediction accuracy and execution time. High-performing models like CBR were found to take the longest execution time. This highlights the need for balanced optimization between model accuracy and computational efficiency.Gradient Boosting vs. Gradient Descent: The study underlined a fundamental difference between gradient boosting and gradient descent approaches, which led to better prediction accuracy with gradient boosting models. This could inspire further research into model architectures and algorithmic design.Role of Regularization in Overfitting Control: The improved performance of the XGB model was attributed to its ability to control overfitting through a more regularized model formalization. This showcases the importance of proper regularization techniques in model development to prevent overfitting.

Table 8 compares different methods used for internet traffic prediction. The Temporal and Spatial Convolutional Network [18] performs with a mean accuracy of 91.05% and uses a single-step forecasting method, but lacks any mechanism for outlier detection or mitigation. On the other hand, the Temporal and Spatial Reinforcement Learning method [31] offers a multi-step forecasting approach with accuracy rates ranging from 90.03% to 97.38% across different time frames and datasets. However, like the Convolutional Network, it does not offer any outlier handling. The Temporal Statistical Technique [32] provides a versatile approach with both single-step and multi-step forecasting. However, its accuracy declines sharply as the forecasting time increases, with only a 53% accuracy at the 10-min mark. It also does not address outlier detection or mitigation. In comparison, our proposed Temporal Boosting Algorithms perform with a consistently high accuracy across various time frames, from 94.92% at 5 min to 92.89% at 60 min. Importantly, this model also includes outlier detection and mitigation, making it a robust and efficient solution for predictive tasks.

## 6. Conclusions

Given the historical traffic volume data of a network segment in the past time intervals, our objective in this study is to predict its future traffic volume in subsequent time intervals. This internet traffic forecasting problem can be divided into two board categories based on forecasting steps: single-step and multi-step prediction tasks. The single-step prediction can forecast one step. In comparison, multi-step forecasting can predict more than one step, which is more beneficial for network capacity planning and improving the quality of service and experience.

In the literature, most of the works deal with single-step forecasting tasks. Also, there is a lack of investigation into single-step and multi-step traffic forecasting tasks with anomalous data, which is common in the real world. Due to many external and internal factors, traffic volume can be abruptly up and down, which does not reflect the general traffic behavior over time. In this study, we integrate the anomaly detection and mitigation module with a machine learning model and experiment with real-world traffic datasets to investigate the impact of outliers on traffic prediction accuracy. We consider five different algorithms from gradient descent and gradient boosting categories for performance analysis. Boosting algorithms are popular for prediction tasks for their higher performance and better generalization capability. They have been used in the existing literature for forecasting tasks, but a comprehensive performance analysis for single-step and multi-step forecasting has not been carried out before. Therefore, we extensively analyze the performance of several state-of-the-art gradient boosting and gradient descent algorithms for single-step and multi-step forecasting based on prediction accuracy and execution time. A total of five different machine learning models, Gradient Boosting Regressor (GBR), eXtreme gradient Boosting (XGB), Light Gradient Boosting Machine (LGB), CatBoost Regressor (CBR), and Stochastic Gradient Descent (SGD), have been considered for our experiment. Among them, the classic CBR model outperforms in single-step and multi-step forecasting, although its execution times were comparatively higher than the rest. After comparing the performance among traditional machine learning models, we integrated a module for anomaly detection and mitigation from our traffic data. As real-world traffic is susceptible to various external and internal factors, the outliers in the data are prevalent. So, the anomaly detection module analyzes the traffic data before feeding them into machine learning models using statistical profiling and unsupervised models. After identifying the abnormal data points, we mitigate them using a method called backward filling, and then we provide them to train our traffic prediction models. As a result, our proposed models outperform the classical machine learning models in terms of prediction accuracy. Again, CBR is the best-performing model with a minimum average gap between actual and predicted traffic. However, CBR prediction accuracy was very close to LGB, while the execution time was significantly higher. Considering both accuracy and execution time, the LGB performance was better among all prediction models.

In the future, we plan to extend our work by integrating signal decomposition techniques such as discrete wavelet decomposition and empirical mode decomposition with prediction. We will use these techniques to separate individual components of our traffic signal and model them using machine learning and deep learning. Also, we would like to include a noise reduction module to remove white noise from traffic data before providing them to our prediction models.

## Figures and Tables

**Figure 1 sensors-24-01871-f001:**
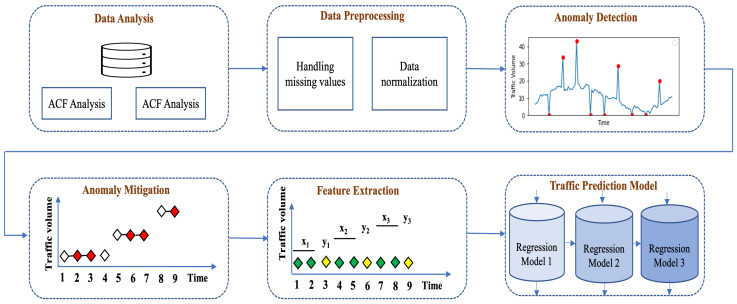
High-level framework of proposed model.

**Figure 2 sensors-24-01871-f002:**
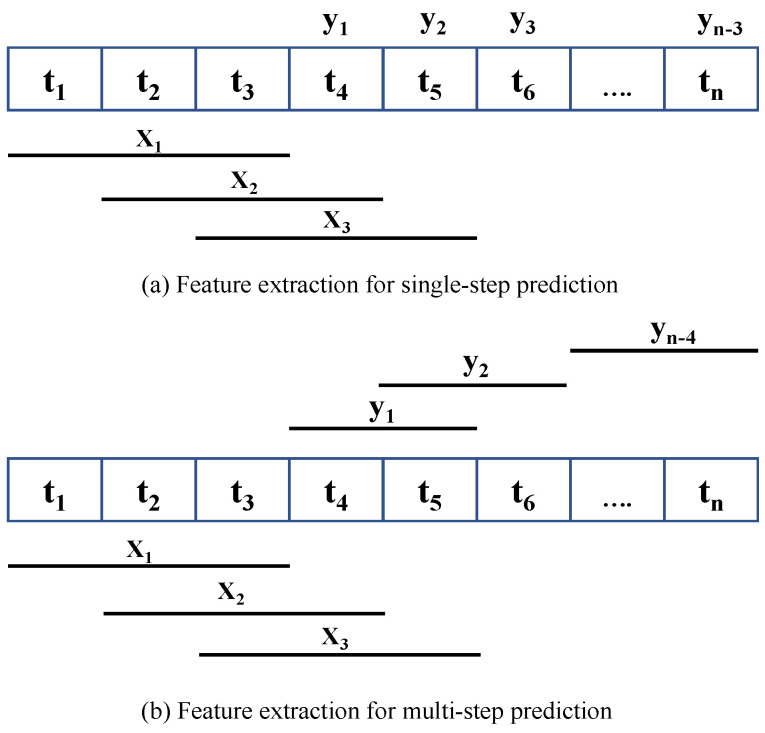
Feature extraction.

**Figure 3 sensors-24-01871-f003:**
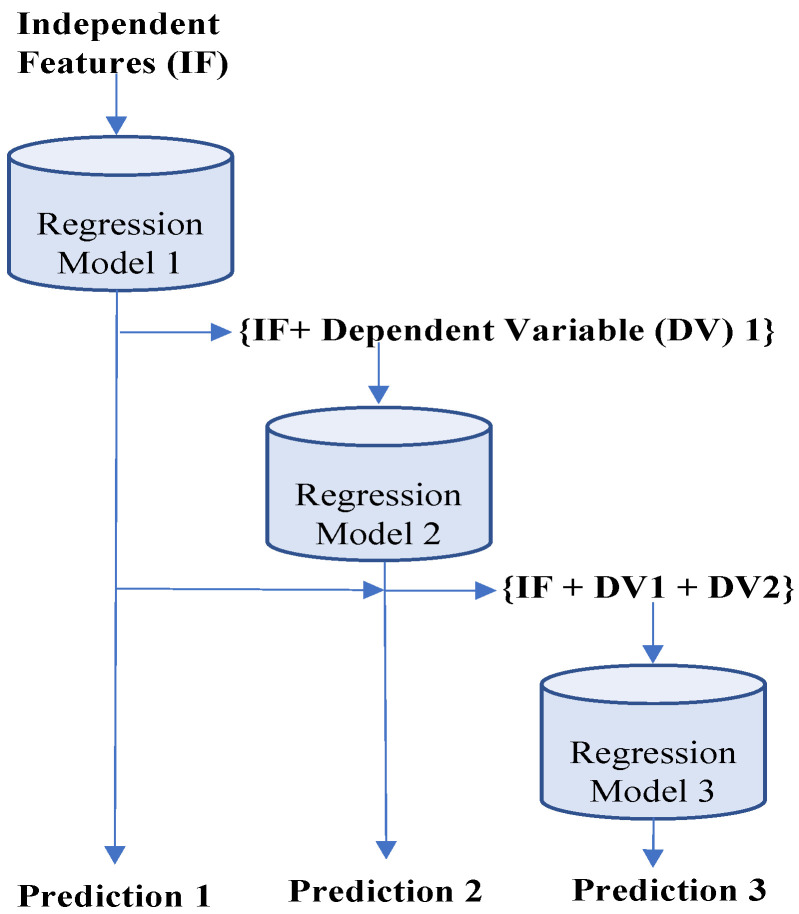
Chain multi-output regression model.

**Figure 4 sensors-24-01871-f004:**
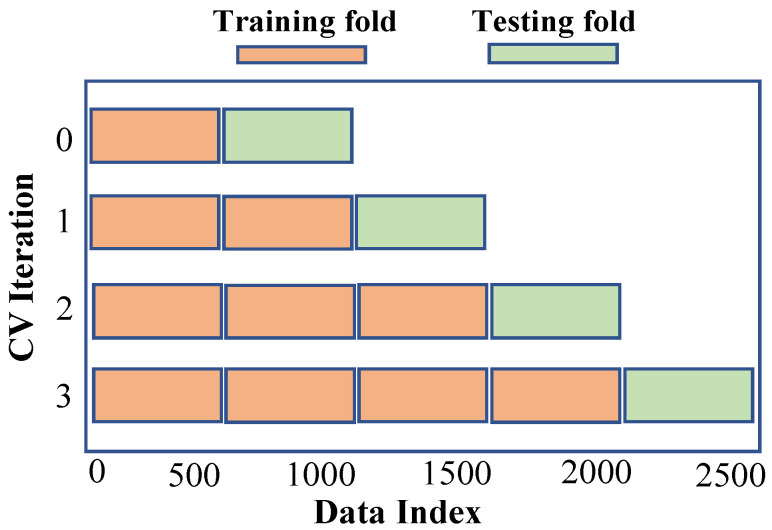
Rolling-based cross-validation.

**Figure 5 sensors-24-01871-f005:**
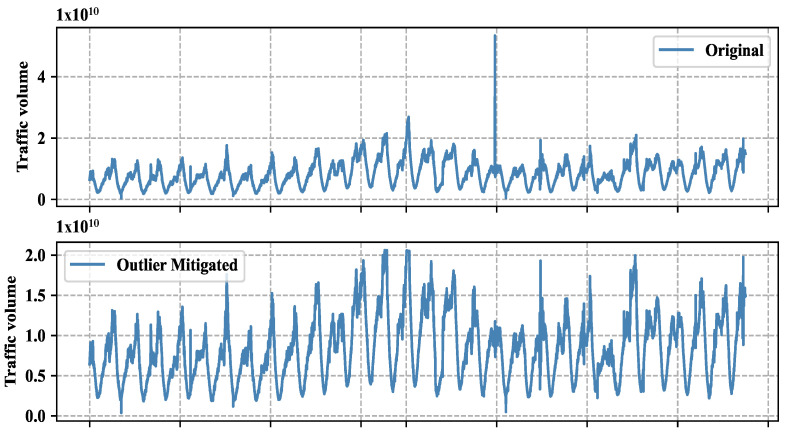
Original traffic vs. outlier mitigated traffic.

**Figure 6 sensors-24-01871-f006:**
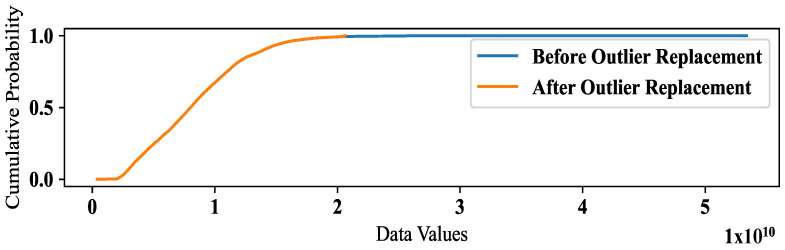
Empirical Cumulative Distribution Function (ECDF) plot of our traffic data.

**Figure 7 sensors-24-01871-f007:**
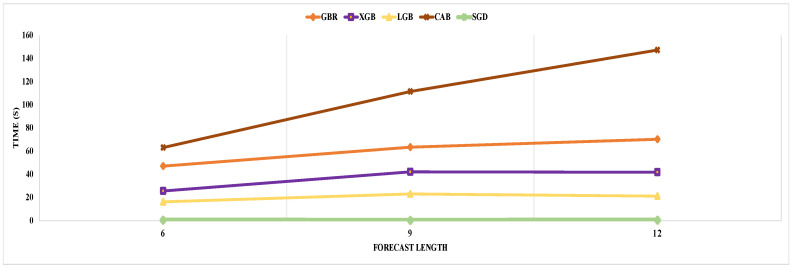
Average execution time of our proposed models for different forecast length.

**Figure 8 sensors-24-01871-f008:**
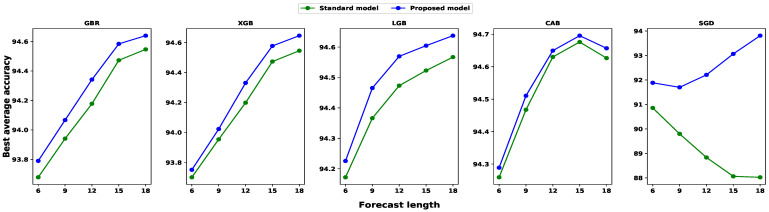
A comparison among different feature subset performances regarding average prediction accuracy for six-step forecasting. The average accuracy is calculated as the mean of each step’s individual prediction accuracy.

**Figure 9 sensors-24-01871-f009:**
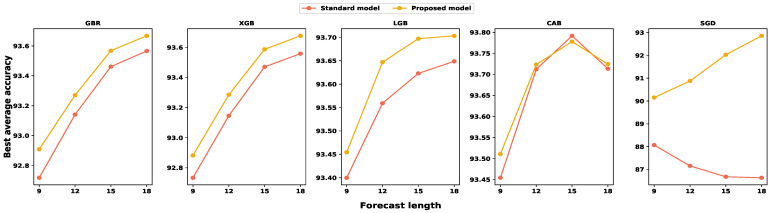
A comparison among different feature subset performances regarding average prediction accuracy for nine-step forecasting. The average accuracy is calculated as the mean of each step’s individual prediction accuracy.

**Figure 10 sensors-24-01871-f010:**
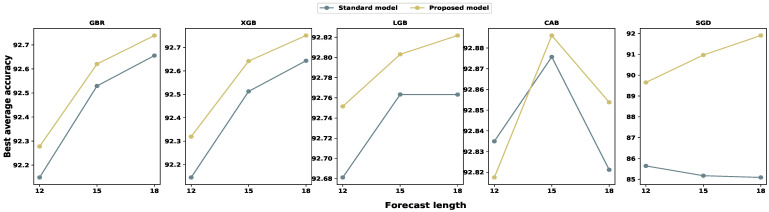
A comparison among different feature subset performances regarding average prediction accuracy for twelve-step forecasting. The average accuracy is calculated as the mean of each step’s individual prediction accuracy.

**Table 3 sensors-24-01871-t003:** Statistical analysis before and after outlier mitigation.

Data Samples Statistics		
Statistics	Before	After
Mean	8,364,742,790	8,344,127,355
SD	4,096,690,529	4,013,117,118
Statistical Test Results Summary		
Test	Test statistic	*p*-value
Two-sample *t*-test	0.328505869	0.742533327
Wilcoxon Rank-sum test	0.015409873	0.987705187
Kolmogorov–Smirnov test	0.005148467	0.999891457

**Table 4 sensors-24-01871-t004:** Single-step forecasting MAPE (%) value for the proposed and standard machine learning models. Each value represents an average gap between actual and predicted traffic volume. The best performance for the corresponding model is highlighted in bold and underlined.

	Input	Classical Model (With Outlier)					Proposed Model (Without Outlier)				
Model		6	9	12	15	18	6	9	12	15	18
GBR		** 7.47 **	7.74	7.57	7.60	7.76	** 5.20 **	5.31	5.24	5.26	5.25
XGB		7.65	7.69	** 7.47 **	7.59	7.60	5.16	5.20	5.19	** 5.15 **	5.17
LGB		8.51	** 8.47 **	8.47	8.47	8.53	5.11	** 5.09 **	5.13	5.14	5.10
CBR		** 7.56 **	7.58	7.64	7.78	8.12	5.10	** 5.08 **	5.20	5.32	5.44
SGD		12.80	10.51	** 10.44 **	11.13	12.16	** 6.10 **	7.27	8.01	8.50	8.23

**Table 5 sensors-24-01871-t005:** Six-step forecasting MAPE (%) value for the proposed and standard machine learning models. Each value represents an average gap between actual and predicted traffic volume for each individual forecast step. The best performance for the corresponding model is highlighted in bold and underlined.

Model		GBR		XGB		LGB		CBR		SGD	
		Classical Model	Prop. Model	Classical Model	Prop. Model	Classical Model	Prop. Model	Classical Model	Prop. Model	Classical Model	Prop. Model
6	Step 1	3.80	3.76	3.78	3.75	3.68	3.63	3.65	3.61	** 6.41 **	5.23
	Step 2	4.95	4.84	4.85	4.85	4.71	4.63	4.55	4.54	** 7.45 **	6.46
	Step 3	5.93	5.85	5.88	5.84	5.53	5.44	5.44	5.38	** 8.62 **	7.57
	Step 4	6.85	6.78	6.84	6.80	6.26	6.22	6.21	6.16	** 9.72 **	8.71
	Step 5	7.74	7.58	7.76	7.63	7.02	6.96	6.90	6.91	** 10.85 **	9.81
	Step 6	8.66	8.46	8.68	8.62	7.77	7.76	7.70	7.68	** 11.80 **	10.92
9	Step 1	3.79	3.70	3.75	3.71	3.67	3.60	3.64	3.60	7.36	5.61
	Step 2	4.82	4.73	4.75	4.71	4.61	4.52	4.55	4.46	8.54	6.71
	Step 3	5.70	5.54	5.71	5.63	5.44	5.31	5.27	5.24	9.76	7.77
	Step 4	6.60	6.42	6.59	6.51	6.08	5.98	5.99	5.94	10.75	8.83
	Step 5	7.31	7.22	7.34	7.21	6.69	6.55	6.56	6.51	11.79	9.92
	Step 6	8.15	7.99	8.14	8.10	7.31	7.25	7.17	7.19	12.99	10.96
12	Step 1	3.92	3.68	3.78	3.69	3.72	3.58	3.60	3.55	8.40	5.38
	Step 2	4.73	4.58	4.62	4.55	4.59	4.43	4.43	4.43	9.49	6.39
	Step 3	5.51	5.36	5.57	5.37	5.31	5.21	5.18	5.12	10.67	7.35
	Step 4	6.29	6.13	6.24	6.13	5.94	5.89	5.74	5.74	11.65	8.31
	Step 5	6.85	6.75	7.00	6.79	6.52	6.45	6.35	6.33	12.86	9.22
	Step 6	7.63	7.45	7.60	7.50	7.08	7.01	6.91	6.92	13.92	10.10
15	Step 1	3.69	3.59	3.71	3.60	3.68	3.60	** 3.66 **	** 3.62 **	9.15	4.87
	Step 2	4.53	4.46	4.53	4.43	4.57	4.43	** 4.44 **	** 4.39 **	10.82	5.73
	Step 3	5.26	5.19	5.26	5.16	5.26	5.14	** 5.14 **	** 5.09 **	11.39	6.59
	Step 4	5.89	5.81	5.91	5.76	5.87	5.79	** 5.66 **	** 5.68 **	12.24	7.36
	Step 5	6.57	6.38	6.59	6.46	6.45	6.41	** 6.27 **	** 6.21 **	13.31	8.15
	Step 6	7.23	7.06	7.16	7.12	7.03	7.01	** 6.76 **	** 6.85 **	14.71	8.90
18	Step 1	** 3.68 **	** 3.55 **	** 3.67 **	** 3.55 **	** 3.60 **	** 3.53 **	3.64	3.62	9.48	** 4.36 **
	Step 2	** 4.45 **	** 4.40 **	** 4.46 **	** 4.39 **	** 4.54 **	** 4.43 **	4.48	4.42	10.50	** 5.12 **
	Step 3	** 5.19 **	** 5.08 **	** 5.17 **	** 5.08 **	** 5.21 **	** 5.09 **	5.12	5.07	11.50	** 5.87 **
	Step 4	** 5.81 **	** 5.73 **	** 5.82 **	** 5.72 **	** 5.81 **	** 5.76 **	5.71	5.67	12.72	** 6.59 **
	Step 5	** 6.51 **	** 6.38 **	** 6.51 **	** 6.39 **	** 6.40 **	** 6.35 **	6.35	6.31	13.34	** 7.25 **
	Step 6	** 7.08 **	** 7.03 **	** 7.10 **	** 6.99 **	** 7.04 **	** 7.02 **	6.93	6.96	14.31	** 7.94 **

**Table 6 sensors-24-01871-t006:** Nine-step forecasting MAPE (%) value for the proposed and standard machine learning models. Each value represents an average gap between actual and predicted traffic volume for each individual forecast step. The best performance for the corresponding model is highlighted in bold and underlined.

Model		GBR		XGB		LGB		CAB		SGD	
		Classical Model	Prop. Model	Classical Model	Prop. Model	Classical Model	Prop. Model	Classical Model	Prop. Model	Classical Model	Prop. Model
9	Step 1	3.78	3.71	3.76	3.69	3.66	3.60	3.67	3.60	** 7.40 **	5.64
	Step 2	4.83	4.67	4.76	4.74	4.61	4.54	4.60	4.52	** 8.56 **	6.73
	Step 3	5.73	5.55	5.69	5.59	5.42	5.33	5.28	5.25	** 9.66 **	7.80
	Step 4	6.69	6.40	6.58	6.42	6.07	6.03	5.98	5.91	** 10.74 **	8.90
	Step 5	7.35	7.20	7.37	7.15	6.65	6.56	6.58	6.53	** 11.90 **	9.90
	Step 6	8.12	7.94	8.10	8.03	7.29	7.24	7.23	7.20	** 13.22 **	10.96
	Step 7	8.97	8.77	9.01	8.79	7.98	7.92	7.96	7.88	** 14.15 **	11.98
	Step 8	9.66	9.43	9.65	9.48	8.55	8.51	8.53	8.50	** 15.49 **	12.91
	Step 9	10.41	10.15	10.49	10.17	9.17	9.17	9.08	9.02	** 16.26 **	13.84
12	Step 1	3.84	3.66	3.82	3.64	3.70	3.58	3.59	3.56	8.47	5.37
	Step 2	4.68	4.58	4.65	4.56	4.57	4.44	4.43	4.41	9.51	6.42
	Step 3	5.53	5.40	5.51	5.39	5.31	5.22	5.18	5.14	10.56	7.36
	Step 4	6.26	6.10	6.20	6.16	5.97	5.84	5.76	5.77	11.74	8.31
	Step 5	6.91	6.77	7.05	6.80	6.57	6.41	6.43	6.36	12.77	9.22
	Step 6	7.63	7.56	7.60	7.52	7.09	6.99	6.96	6.95	13.97	10.12
	Step 7	8.28	8.16	8.34	8.18	7.73	7.69	7.57	7.53	14.92	10.97
	Step 8	8.98	8.84	8.96	8.78	8.24	8.25	8.08	8.06	16.03	11.74
	Step 9	9.64	9.50	9.56	9.40	8.78	8.77	8.60	8.69	17.62	12.59
15	Step 1	3.73	3.59	3.68	3.57	3.64	3.58	** 3.67 **	** 3.64 **	9.12	4.88
	Step 2	4.57	4.42	4.52	4.43	4.54	4.42	** 4.51 **	** 4.41 **	10.37	5.74
	Step 3	5.26	5.16	5.23	5.15	5.27	5.15	** 5.10 **	** 5.08 **	11.28	6.55
	Step 4	5.89	5.86	5.90	5.84	5.86	5.77	** 5.69 **	** 5.67 **	12.28	7.36
	Step 5	6.60	6.46	6.63	6.48	6.46	6.41	** 6.28 **	** 6.23 **	13.36	8.19
	Step 6	7.22	7.15	7.19	7.10	7.09	6.99	** 6.76 **	** 6.90 **	14.46	8.90
	Step 7	7.89	7.77	7.91	7.77	7.71	7.63	** 7.40 **	** 7.42 **	15.44	9.37
	Step 8	8.54	8.41	8.52	8.39	8.09	8.13	** 7.93 **	** 8.04 **	16.33	10.05
	Step 9	9.14	9.07	9.19	8.99	8.73	8.65	** 8.53 **	** 8.61 **	17.24	10.72
18	Step 1	** 3.66 **	** 3.54 **	** 3.68 **	** 3.54 **	** 3.61 **	** 3.55 **	3.61	3.63	9.68	** 4.36 **
	Step 2	** 4.45 **	** 4.43 **	** 4.46 **	** 4.39 **	** 4.52 **	** 4.43 **	4.49	4.44	10.60	** 5.12 **
	Step 3	** 5.19 **	** 5.06 **	** 5.19 **	** 5.10 **	** 5.21 **	** 5.11 **	5.13	5.09	11.39	** 5.87 **
	Step 4	** 5.79 **	** 5.76 **	** 5.84 **	** 5.74 **	** 5.85 **	** 5.75 **	5.73	5.66	12.42	** 6.59 **
	Step 5	** 6.59 **	** 6.40 **	** 6.52 **	** 6.37 **	** 6.47 **	** 6.35 **	6.33	6.28	13.34	** 7.27 **
	Step 6	** 7.12 **	** 7.00 **	** 7.10 **	** 6.97 **	** 6.98 **	** 7.03 **	6.95	6.94	14.40	** 7.91 **
	Step 7	** 7.80 **	** 7.66 **	** 7.88 **	** 7.67 **	** 7.63 **	** 7.57 **	7.52	7.52	15.24	** 8.43 **
	Step 8	** 8.33 **	** 8.27 **	** 8.34 **	** 8.24 **	** 8.15 **	** 8.16 **	8.11	8.16	16.12	** 9.06 **
	Step 9	** 8.98 **	** 8.88 **	** 8.97 **	** 8.90 **	** 8.74 **	** 8.72 **	8.70	8.75	17.10	** 9.69 **

**Table 7 sensors-24-01871-t007:** Twelve-step forecasting MAPE (%) value for the proposed and standard machine learning models. Each value represents an average gap between actual and predicted traffic volume for each individual forecast step. The best performance for the corresponding model is highlighted in bold and underlined.

Model		GBR		XGB		LGB		CBR		SGD	
		Classical Model	Prop. Model	Classical Model	Prop. Model	Classical Model	Prop. Model	Classical Model	Prop. Model	Classical Model	Prop. Model
12	Step 1	3.92	3.67	3.82	3.65	3.70	3.58	3.59	3.56	** 8.42 **	5.36
	Step 2	4.75	4.57	4.69	4.54	4.56	4.46	4.44	4.42	** 9.47 **	6.40
	Step 3	5.52	5.38	5.52	5.42	5.30	5.23	5.17	5.12	** 10.59 **	7.36
	Step 4	6.24	6.11	6.26	6.11	5.92	5.85	5.77	5.76	** 11.75 **	8.35
	Step 5	6.91	6.80	7.00	6.84	6.55	6.47	6.42	6.34	** 12.85 **	9.23
	Step 6	7.62	7.48	7.61	7.41	7.11	6.95	6.94	6.99	** 13.82 **	10.15
	Step 7	8.26	8.16	8.34	8.17	7.73	7.66	7.50	7.55	** 15.05 **	11.02
	Step 8	8.94	8.80	8.94	8.71	8.31	8.21	8.05	8.11	** 15.91 **	11.78
	Step 9	9.56	9.50	9.56	9.40	8.84	8.84	8.63	8.74	** 17.03 **	12.58
	Step 10	10.22	10.11	10.20	10.03	9.35	9.31	9.18	9.25	** 18.15 **	13.41
	Step 11	10.81	10.68	10.88	10.63	9.96	9.90	9.79	9.87	** 19.16 **	13.86
	Step 12	11.46	11.39	11.43	11.27	10.52	10.52	10.50	10.48	** 20.11 **	14.69
15	Step 1	3.68	3.61	3.70	3.60	3.66	3.57	** 3.68 **	** 3.63 **	9.12	4.84
	Step 2	4.57	4.48	4.52	4.44	4.54	4.43	** 4.51 **	** 4.42 **	10.23	5.74
	Step 3	5.28	5.23	5.29	5.15	5.22	5.16	** 5.11 **	** 5.10 **	11.32	6.61
	Step 4	5.99	5.81	5.95	5.83	5.84	5.75	** 5.73 **	** 5.70 **	12.50	7.44
	Step 5	6.55	6.50	6.62	6.48	6.45	6.39	** 6.28 **	** 6.19 **	13.43	8.19
	Step 6	7.23	7.12	7.25	7.03	7.09	7.02	** 6.84 **	** 6.82 **	14.30	8.91
	Step 7	7.86	7.73	7.91	7.81	7.74	7.66	** 7.48 **	** 7.45 **	15.37	9.61
	Step 8	8.51	8.41	8.50	8.39	8.08	8.14	** 7.99 **	** 7.99 **	16.46	10.03
	Step 9	9.16	9.03	9.14	9.01	8.73	8.72	** 8.53 **	** 8.62 **	17.36	10.71
	Step 10	9.65	9.66	9.72	9.64	9.25	9.26	** 9.16 **	** 9.17 **	18.32	11.43
	Step 11	10.26	10.19	10.30	10.14	9.80	9.86	** 9.82 **	** 9.80 **	19.24	12.13
	Step 12	10.90	10.79	10.94	10.78	10.43	10.40	** 10.36 **	** 10.47 **	20.27	12.80
18	Step 1	** 3.68 **	** 3.55 **	** 3.65 **	** 3.53 **	** 3.63 **	** 3.53 **	3.68	3.61	9.51	** 4.35 **
	Step 2	** 4.46 **	** 4.41 **	** 4.46 **	** 4.38 **	** 4.58 **	** 4.42 **	4.51	4.40	10.43	** 5.15 **
	Step 3	** 5.20 **	** 5.10 **	** 5.20 **	** 5.09 **	** 5.21 **	** 5.12 **	5.16	5.10	11.48	** 5.87 **
	Step 4	** 5.84 **	** 5.74 **	** 5.83 **	** 5.74 **	** 5.91 **	** 5.73 **	5.76	5.69	12.43	** 6.57 **
	Step 5	** 6.49 **	** 6.40 **	** 6.57 **	** 6.40 **	** 6.47 **	** 6.36 **	6.36	6.28	13.66	** 7.27 **
	Step 6	** 7.12 **	** 6.97 **	** 7.15 **	** 6.99 **	** 7.06 **	** 7.01 **	6.93	6.95	14.47	** 7.96 **
	Step 7	** 7.84 **	** 7.69 **	** 7.80 **	** 7.66 **	** 7.76 **	** 7.67 **	7.58	7.52	15.32	** 8.47 **
	Step 8	** 8.31 **	** 8.27 **	** 8.35 **	** 8.21 **	** 8.19 **	** 8.10 **	8.14	8.15	16.14	** 9.04 **
	Step 9	** 8.96 **	** 8.97 **	** 9.02 **	** 8.92 **	** 8.70 **	** 8.74 **	8.68	8.73	17.18	** 9.72 **
	Step 10	** 9.49 **	** 9.42 **	** 9.45 **	** 9.43 **	** 9.20 **	** 9.28 **	9.18	9.19	18.20	** 10.31 **
	Step 11	** 10.03 **	** 10.02 **	** 10.06 **	** 10.06 **	** 9.78 **	** 9.81 **	9.74	9.77	19.82	** 10.89 **
	Step 12	** 10.71 **	** 10.60 **	** 10.72 **	** 10.57 **	** 10.35 **	** 10.36 **	10.42	10.38	20.28	** 11.58 **

**Table 8 sensors-24-01871-t008:** A performance comparison of our proposed method compared to existing methods.

Ref.	Dataset	Method	Forecast Length	Outlier Detection and Mitigation	Mean Accuracy
[18]	temporal and spatial	convolutional neural network	single-step	no	91.05%
[31]	temporal and spatial	reinforcement learning	multi-step	no	93.12% (15 min), 91.68% (30 min), 90.03% (60 min)
[32]	temporal	statistical technique	single-step and multi-step	no	90% (1 min), 70% (5 min), 53% (10 min)
Proposed	temporal	boosting algorithms	single-step and multi-step	yes	4.92% (5 min), 94.70% (30 min), 93.78% (45 min), 92.89% (60 min)

## Data Availability

Data are contained within the article.

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
