# Peer review of "Multi-Step Internet Traffic Forecasting Models with Variable Forecast Horizons for Proactive Network Management [Author-notes fn1-sensors-24-01871]"

_sensors, 2024, doi:10.3390/s24061871_

Round 1

Reviewer 1 Report

Comments and Suggestions for Authors

This paper proposes an efficient ITF framework. However, I suggest optimizing the paper's writing, especially in terms of logical coherence, to effectively convey the innovative aspects to the readers.

1. The paper argues that "Effective ITF frameworks are necessary to manage these networks and prevent network congestion and over-provisioning," and the paper proposes an effective ITF framework. However, it is important to describe the key differences or contributions of this framework compared to existing ones, as there is already a large amount of effective TTF frameworks available.

2. Multi-step prediction is a common and widespread approach, and there are already many models that have implemented multi-step prediction. I suggest conducting a more in-depth literature review and refining the discussion and viewpoints accordingly.

3. The formatting of some figures needs improvement, as there are compression in figures such as Figure 4 and Figure 5.

4. It would be beneficial to include experiments with deep learning models in both single-step and multi-step prediction tasks. Deep learning models are generally considered to perform better in most prediction tasks, especially with novel deep learning architectures. To my knowledge, there are already some multi-task deep learning models that can simultaneously handle missing data imputation and prediction tasks.

Comments on the Quality of English Language

1. I recommend revising the abstract section. Typically, it is not necessary to list the comparative methods one by one in the abstract. Additionally, the logical coherence of the last sentence in the abstract needs improvement.

2. The research significance of the paper needs to be further clarified, as the current expressions in the abstract and subsequent sections are not entirely consistent. The mention in the abstract that "it is challenging for predicting traffic volume in large-scale networks" gives the impression that you are addressing traffic volume prediction in large-scale networks. However, the subsequent introduction does not have a strong connection to large-scale networks.

Reviewer 2 Report

Comments and Suggestions for Authors

This study proposes multi-step forecasting of wireless network traffic leveraging various ML algorithms, with a particular focus on ensemble methods. The paper is well-structured and written. However, I have the following comments to further improve certain aspects:

1. Why are the ML algorithms limited to those used in this study? What about time-series prediction algorithms such as LSTM autoencoders? Multi-step prediction is particularly well-suited for time-series prediction.

2. Most papers start with Section 1 as the introduction, not Section 0. I recommend that the authors include a table in Section 1 (related work) that uses O or X to identify some important aspects of each study, indicate whether synthetic or real-world datasets are used, and identify the distinction between outliers, etc.

3. In Section 1, the authors mention that outliers are frequent and complicate prediction. However, these outliers are often actual network events or traffic, which are unavoidable. Is the approach of simply removing or replacing such outliers the correct choice? Perhaps for prediction accuracy, it might be beneficial, but it could lead to missing real-world network events.

4. Many studies compare ML algorithms, secure some generalizability, and automatically formulate algorithms. Please cite and compare or discuss these in relation to this study:

- "Machine Learning-Based Prediction Models for Control Traffic in SDN Systems." IEEE Transactions on Services Computing (2023).

- "Machine-learning based prediction of multiple types of network traffic." International Conference on Computational Science. Cham: Springer International Publishing, 2021.

- "Probabilistic regularized extreme learning for robust modeling of traffic flow forecasting." IEEE Transactions on Neural Networks and Learning Systems (2020).

- "Robust genetic machine learning ensemble model for intrusion detection in network traffic." Scientific Reports 13.1 (2023): 17227.

- "Control Channel Isolation in SDN Virtualization: A Machine Learning Approach," IEEE/ACM 23rd International Symposium on Cluster, Cloud, and Internet Computing (2023).

Comments on the Quality of English Language

NA

Round 2

Reviewer 1 Report

Comments and Suggestions for Authors

The quality of the paper has been improved by revising it and now there is one more piece of advice.

1. In the Methods section it would be a good idea to use an aesthetically pleasing and well-organized picture to show the overall framework of the model in this paper.

Comments on the Quality of English Language

No recommendations

Reviewer 2 Report

Comments and Suggestions for Authors

Many of my previous review concerns have been solved with the authors' efforts. However, I still have a concern about the limit on applied models in that the used algorithms are traditional, and other SOTA algorithms are already used in existing studies.

Comments on the Quality of English Language

NA
